# Infodemic and Fake News in Spain during the COVID-19 Pandemic

**DOI:** 10.3390/ijerph18041781

**Published:** 2021-02-12

**Authors:** María Jesús Fernández-Torres, Ana Almansa-Martínez, Rocío Chamizo-Sánchez

**Affiliations:** Department of Audiovisual Communication and Advertising, Faculty of Communication Sciences, Universidad de Málaga, 29071 Malaga, Spain; aam@uma.es (A.A.-M.); rcs@uma.es (R.C.-S.)

**Keywords:** fake news, infodemic, COVID-19, social media, health crisis, health communication, public opinion

## Abstract

Internet, new technologies and social networks have changed the consumption and dissemination of information. The world is witnessing the proliferation of so-called false news, especially since the beginning of 2020, when COVID-19 became the main issue on the global agenda. Alleged government actions, remedies, advice, etc., are the cause of a multitude of messages that are often false. Through surveys (1115 responses were obtained) and a review of the literature, we explore how the proliferation of COVID-19’s false news affects and impacts public opinion in Spain. We also examine how citizens are being informed about the pandemic, identify the main channels of communication used and discover the impact of misinformation. The main conclusions are that, in Spain, citizens are interested in information related to the coronavirus, but there is a lack of media credibility and reliability; the social networks and instant messaging are considered the channels that transmit the greatest amount of false news.

## 1. Introduction

During the coronavirus health crisis, the term infodemic has become popular ([1] p. 7). It refers to information about the pandemic that has been experienced, coinciding with the health crisis. The WHO [2] warned that much of the information circulating was often false. “We are facing a pandemic because of the overabundance of information, which makes it difficult for some people to find reliable resources or trusted guides when they need them” ([3] p. 5).

Hoaxes or fake news are not a new phenomenon [4]. We can go back to stories told by minstrels in the Middle Ages of people dying in battle and then being resurrected. The fabrication of false news of Nazis and Soviets or, more recently, the invention of weapons of mass destruction to justify participation in the Iraq war, are other examples of false news throughout history.

In other words, there has always been fake news, but in cyber society a double phenomenon has occurred, which makes hoaxes circulate at great speed. On one hand, the evolution and implementation of technologies make tools available to almost everyone to distribute information throughout the world [5,6]. On the other hand, as a consequence of this, the number of senders is increasing. It is no longer only the classic political actors (political parties, institutions, media) who disseminate information. Today, practically anyone can do it. Citizens have gone from being mere consumers of content to being prosumers [7]. The digital agora was born, in which the so-called intelligent multitudes participate [8]. 

This obviously has very positive consequences in terms of citizen participation, which is crucial in a democracy. However, it also has negative consequences in terms of the distribution of fake news. “Incorrect retention and false attribution contribute to the biggest problems that authorities face in the multiplatform scenario: how to deal with rumours and fake news” ([9] p. 11). During the pandemic, “digital media emerged as the first option for information, followed by TV news, social networks and instant messaging applications” ( [1] p. 5).

In fact, “the volume of news in the digital media in Spain has increased considerably since the state of alert was decreed” ([10] p. 7). This is a trend that has continued throughout the pandemic [11]. 

For a fake news story to be successful, it is enough for the subject to attract attention. The health crisis is a matter of maximum interest, so it was foreseeable that it would become a breeding ground for lies and hoaxes, an environment conducive to the development of false news. Moreover, it is a highly emotionally charged issue [12], so at times when political communication puts the emotional before the rational, the success of false news is guaranteed.

Likewise, a situation of crisis, of uncertainty, is a good environment for creating fake news, with the political and social intention of benefiting or harming groups, with economic intentions or simply for the pleasure of doing evil [13]. “The premeditated use of false information for the benefit of organizations continues today through the use of false news opinions or facts that contribute to the improvement of reputation or the destruction of that of competitors” ([4] p. 1725). 

It coincides in a context where there is a problem of credibility towards the power (political parties and public institutions) and towards the media. The disaffection [14] of citizens towards politicians has been a constant for decades. To this situation we must add the lack of credibility of the media. The professional crisis of journalism and its precariousness, as well as the economic and ideological interests of the media companies, have caused this lack of credibility. “The public maintains a critical attitude towards the information provided by the media, which they consider to be conditioned by their editorial line, and they do so in a sensationalist way” ([1] p. 9). 

Moreover, the media are no longer the indispensable mediators between power and citizens, since Technology of the Information and Communication offer other communication tools that can make them dispensable. The importance of social media is increasing, to the detriment of traditional mass media [15,16]. Furthermore, another added problem is the lack of scientific data and real knowledge of the virus. In the absence of definitive evidence to refute false news and rumours, the fight against the pandemic is much more difficult [17].

In this context, it was not difficult for fake news to proliferate—fake news that, before the health crisis, was a concern for many political institutions. In this sense, the initiative that took place in Europe, where a group of experts analysed the risks of the phenomenon and produced a report [18], stands out. The experts insist on the need to disseminate manuals or codes of good practice, and feel that training citizens, media and digital literacy, and the creation of critical awareness are the only ways to combat the infodemic [19,20].

On the other hand, in any crisis situation, so that lies and rumours do not thrive, communication must be strengthened, through proper management of external and internal communication tools. Communication is the antidote to the spread of rumours and falsehoods and allows political actors to be seen as reliable and credible sources [21,22,23]. As Xifra [24] points out, a crisis “is a time for public relations and reputational risk management, rather than advertising” (p. 12). The harder the crisis and its consequences, the more damage can be done to the image of the organizations that are held accountable in that crisis [25,26]. Therefore, it is necessary to have reinforced and fortified communication, especially in crisis situations, where communication management is of vital importance so that said crisis does not have a negative impact on the image or, failing that, minimize its impact or leave reinforced.

It is therefore of great interest to analyse the institutional communication carried out by the Spanish government, the opposition parties and the media [27,28,29]. During the state of alarm, the Spanish government gave daily press conferences, sometimes up to three on the same day. In turn, the Ministry of Health’s Centre for the Coordination of Health Alerts and Emergencies held at least one press conference a day. Furthermore, the President of the Government of Spain Pedro Sanchez gave one or two press conferences every week. In total, 71 government press conferences were held during the lockdown, with 414 media outlets participating, which asked 1069 questions, according to Castillo-Esparcia, Fernández-Souto and Puentes-Rivera [30]. 

On the other hand, updated information was offered to citizens on institutional websites and profiles on social networks. Specifically, 1080 publications were posted to the official accounts of La Moncloa on Facebook and Twitter [30]. 

“The crisis resulting from the spread of the COVID-19 is unprecedented, and breaks with the models of communicative crisis management” ([24] p. 3). Therefore, as the author indicates, the management is very complex. 

“The characteristics of institutional communication management have been, firstly, the permanent presence of the government to provide information. At all times, a strategy has been followed to control the topics and the frames” ([30] p. 18). The use of a language of war, associated with national unity, is also noteworthy. Expressions such as “the enemy is the virus”, “we are at war with the virus” or “we will win the battle” were repeated throughout the state of alarm.

The communication strategy has been media-based, with no real interaction with citizens. The press in Spain, as well as in other countries, has been attended to. The health crisis in Spain has been followed by many international media. During the state of alarm, the foreign press published up to 778 news items about COVID-19 in Spain, with a prevalence in the US press and on economic rather than health-related issues [30]. The media strategy can also be seen in social networks, where the contents of press conferences have been reproduced and their potential as a direct communication channel with citizens has largely been wasted [30].

In response to the avalanche of fake news, the National Police issued a guide to prevent citizens from being manipulated by false information [31,32]; it compiled the main false news that had been disseminated. In this regard, the work of organizations such as Maldito Bulo [33] and Newtral [34], which throughout the pandemic have denied false news, is noteworthy.

On the other hand, it is also interesting to analyse the behaviour of the opposition parties, which undoubtedly has consequences on the information that reaches the citizens. Crespo and Garrido [35] have studied this position and consider that “unlike other countries, in Spain, there has not been a complete alignment of the political forces behind the government, typical of the closing of ranks or rally round the flag effect that is characteristic of major crises, but rather there has been a blame game about political responsibilities for the crisis, in which each political force has tried to create its own framework and communication strategy” ([35] p. 17).

For their part, the mass media must play a decisive role in the dissemination of (accurate) information and stop the dissemination of false or erroneous information in crisis situations [36,37], because the public demands such accurate information [38,39]. However, according to studies carried out on the pandemic in Spain, “the media are conditioned by their editorial line when it comes to reporting on the coronavirus. This presence of ideological bias (…) can constitute a bridge to disinformation” ([1] p. 9), which generates uncertainty and lack of confidence in the media [40,41,42,43,44]. 

For all the above reasons, this study aims to find out how the proliferation of fake news on COVID-19 affects citizens and the impact it has on public opinion. This research is limited to the period of the state of alarm decreed in Spain in March 2020, which lasted more than three months, specifically from 14 March to 21 June of that year.

The secondary objectives of the research are: to find out the main ways in which the public has been informed about the pandemic; to identify those sources that give it greater credibility; to find out about the assessment of the institutional information; to identify the content of the majority of the hoaxes received as well as the dissemination that they have had. 

The objectives that have been set out are intended to provide answers to a series of research questions:Q1.What follow up has been done on the information related to COVID-19 and what were the main routes used?Q2.Which sources are the most reliable for citizens and therefore those that generate the least amount of fake news?Q3.What content is most widely disseminated as fake and what are their intentions?Q4.What is the public’s assessment of the institutional information provided by central, regional and local governments about the COVID-19?Q5.What negative effects and consequences for society can fake news have and is it considered necessary to regulate this legally?

## 2. Materials and Methods 

This research is descriptive and has a quantitative methodology, centred on the survey, which, as Buendía, Colás and Hernández point out [45], responds to problems both in descriptive terms and in the relationship of variables thanks to the systematic collection of information. The survey is complemented by an intense bibliographic and newspaper review for the construction of the theoretical framework.

The questionnaire is composed of a total of 22 questions, in addition to the initial discriminatory issues relating to demographic variables (age, gender, training and employment status). The main objective of the questionnaire was to find out how Spanish citizens inform themselves about everything related to the COVID-19, to identify the main channels of communication they use and to discover the repercussions that misinformation can have. The aim was also to find out how Spanish society views the information provided by central, regional and local governments about the pandemic; the way in which citizens have disseminated fake news; and the need (or lack thereof) for a legal framework to regulate false news.

The survey was answered by a total of 1115 people and was active for two months. It was launched on 6 April 2020 and closed on 8 June of the same year. It coincided with the period of home confinement and the period of the toughest restrictions. The survey was carried out in Google Forms software and the sample was random, covering all of Spain. It was distributed online, especially through social networks and instant messaging such as WhatsApp. 

The survey had a margin of error ± 3 and a confidence level of 95%. The reliability of the survey is thus evident. 

The questionnaire was answered by 63% women and 37% men. In terms of the age of the respondents, the vast majority were in the ranges of 35–44 and 45–54 years of age, both adding up to 53%. This was followed by those aged 25–34 (16%) and 55–65 (13%). To a lesser extent, the survey was conducted on people aged 20–24 (8%), over 65 (6%) and 14–19 (4%).

Most of the respondents had a high level of education. Forty-five percent had a higher education degree and 37% had a postgraduate degree. Sixteen percent had secondary education and 2% had basic training. 

Most of the participants, 72%, were working. Students accounted for 12%, and retirees and the unemployed represented 8% and 7%, respectively. Housewives accounted for 1%. 

Twenty-eight percent of those surveyed were public employees, a similar percentage to that of employed workers (27%). This was followed by self-employed workers (14%). The remaining 3% were expatriates and paid trainees.

After processing the responses, the data were analysed to extract results. The data analysis has been carried out through statistical calculations. The analysis of results has led to the conclusions included in this article.

## 3. Results

Practically all of those surveyed said that they had followed the information on the COVID-19 crisis. As can be seen in Figure 1, television has been the medium through which citizens have been most informed, with figures very similar to those of the online press (88%). It is followed by WhatsApp (62%) and the social network Facebook (53%). Radio is used to find out about the pandemic by 46% of those surveyed and 44% of them go to social networking sites to find out about it. Among the latter, Twitter (36%), Instagram (23%) and YouTube (20%) should be highlighted. Twenty percent use the conventional press to find out about the pandemic. Sixteen percent use email and only 5% use other types of instant messaging.

The sources that generate the most credibility, as indicated in Figure 2, are the online press (50%) followed by television (49%) and the institutional websites themselves (47%). Radio is valued as a credible medium by 36%, followed very closely by the profiles that the institutions have on social networks (33%). The printed press is valued positively by 22% of those surveyed. It is striking that the rest of the media is only well-rated for receiving reliable information by less than 8%. This group includes the social network Twitter (7%), webs/blogs (5%), WhatsApp (4%), Facebook and YouTube (3% each); while email, Instagram and other types of messaging are credible to only 2% of respondents.

It is surprising that a large proportion of respondents use WhatsApp and Facebook to find out about COVID-19 when there are other sources that generate a greater degree of credibility. With regard to the follow up that citizens have carried out on institutional information, 93% have followed it in various ways. Seventy-seven percent stated that they had done so through press conferences in which the latest data related to the pandemic were offered; the appearances of technicians and institutional representatives were followed by 67% of those surveyed. Presidency press conferences were also chosen by 63% of participants, while 57% preferred the information provided through public appearances and/or presidential speeches. Only 7% of the participants did not follow up on institutional information. Forty-seven percent of those who stated that they had followed the institutional information in order to obtain documentation valued this official information positively, considering it sufficient and true. Forty-two percent expressed their dissatisfaction with the institutional information provided, considering it insufficient. The remaining 12% do not know or preferred not to answer.

As can be seen from Figure 3, participants’ assessments of the information provided by different governments and administrations on the health crisis differed from one government to another. Thus, 43% of those surveyed gave a positive assessment of the information from central government on COVID-19 to which they have had access, compared with 26% who rated it negatively. Thirty percent of people who have had access to the information provided by the government classified it as mediocre and the remaining 2% did not know or did not answer. 

With regard to the information provided by the territorial, regional and autonomous governments, 37% valued it positively while 18% thought the opposite, qualifying it negatively. Forty percent of those surveyed described the information on COVID-19 offered by these administrations as mediocre. Five percent did not know or did not answer.

With regard to local governments and the information on the coronavirus health crisis, 32% considered it to be positive; 23% considered it to be negative and 32% considered that the information from local governments on COVID-19 has been mediocre.

In other hand, 92% of people surveyed have detected COVID-19-related hoaxes on their own. False news has reached them in many different ways, as can be seen in Figure 4, with WhatsApp standing out, as indicated by 86% of the participants, followed by the social networks Facebook (58%) and Twitter (31%). Other media pointed out by the participants, although to a lesser extent than the previous ones, were: online press (23%), Instagram (19%), television (17%), websites and blogs (15%) and YouTube (14%). The media through which they indicated they have received the least amount of false news were: email (6%), profiles of government institutions (5%, standing out as the most reliable media when considering the channels through which they have received the least amount of false news), the printed press, radio and the websites of government institutions (4%).

The intention of the COVID-19 hoaxes, of which the respondents were aware, was negative for 79% of the participants. Only 8% considered that the news items that were lacking in content showed a positive intention and 13% classified it as neutral.

With regard to the fake news that they remember receiving, 66% of the respondents gave information on how to avoid infection, followed closely by content related to supposed home methods for finding out if a person is infected by a coronavirus and supposed statements by health professionals and security personnel, both of which were expressed by 60% of those surveyed. Information on how the virus is transmitted was highlighted by 57%, and 49% were given lies about supposed medications to be taken or avoided in the case of having symptoms related to the virus. Other notable fake news referred to supposed information on the preventive measures taken in the country in the face of the crisis (massive fumigations, total suspension of classes and exams …) highlighted by 42% of participants; the impersonation of official sources, supposed data on the increase in infected people, supposed declarations by political leaders and supposed information on the protocols to follow during the crisis in the face of basic actions, such as going to the supermarket, were pointed out by approximately 30% of those surveyed. Twenty-one percent of participants reported fake news information about known people who had allegedly become infected and information about what those known infected people allegedly did while they had the disease.

The content of the hoaxes that have been pointed out to a lesser extent by respondents has been related to supposed information about the protocols to be followed during the crisis in basic situations, such as requesting medical attention. Of note in this respect are false reports whose content is related to the affliction of the virus in pets, audios of people impersonating healthcare personnel, conspiracy theories, the appearance of new viruses or false information about healthcare material.

Twenty-three percent of respondents unconsciously shared some false news about COVID-19, believing it to be true and later realising that it was a hoax. In this sense, the majority of participants, 70%, have not shared false news and 7% did not know or preferred not to respond. Along these lines, 92% of those surveyed did not share any kind of fake news (audio, video or information) about COVID-19 because they understood it was a hoax and 96% did not share it, knowing for sure that it was a false news item. It should be noted here that 2% of those surveyed (22 people) shared fake news despite knowing that they were, citing multiple reasons, including joking, generating fear or uncertainty, to observe the reactions of the recipients to such hoaxes or for discrediting an institution, person, group or entity. The other 2% preferred not to answer.

Figure 5 shows the media that people participating in the survey believed were most likely to generate the largest amount of false news about COVID-19. Social networks stand out in the first position, with 92% of responses. They are followed by instant messaging such as WhatsApp, with 81%. In third position, and far behind the previous ones, is the online press, with 26%. Seventeen percent of respondents cite television as the fourth most popular medium for generating fake news. News agencies (6%), radio and print media (5% each) were indicated by respondents as less likely to be used for hoaxes.

With regard to the impact of fake news on COVID-19 in society, Figure 6 shows that the majority of participants, 89%, considered them to be transcendental and dangerous for society. Thus, 49% classified them as “severe” and 40% as “very severe”. Ten percent of the participants believed that the scope of the hoaxes on the coronavirus was moderate and only one percent believed that they had mild consequences for citizens as a whole.

As regards the negative effects that the respondents believe it may have caused in the population, the first thing that stood out was the social alarm (88%), followed by a harmful impact on the reputation of the central government and the autonomous or local governments (50%). The negative effect that hoaxes have on the reputation of a political party was expressed by 44%. Other harmful effects were reported by 30% of respondents, in this order: the reputation of an organization or company, the reputation of a group, the reputation of an individual and financial losses.

Ignorance and damage to the image and reputation of individuals and organizations were the main causes pointed out by respondents for the transmission of coronavirus hoaxes. Sixty percent of participants indicated them as the main reasons. Another noteworthy finding is 51% of respondents said that hoaxes are spread because those who believe them think that they can benefit personally or in some way benefit the group with which they sympathise. Thirty-two percent highlighted that false news was disseminated simply to generate content.

Sixty-four percent of the respondents, 714 people, considered that they were well-informed about COVID-19. Twenty-five percent (279 people) said the opposite: they were not sufficiently informed about the health crisis. Eleven percent of respondents (123 people) preferred not to answer. 

With regard to the legal regulation of fake news, the vast majority of staff taking part in the survey (88%) believed that it is necessary to regulate it, as opposed to 7% who did not consider it essential. Five percent of those surveyed did not know or preferred not to answer.

## 4. Conclusions

This research provides novel contributions in relation to infodemics in times of pandemic in Spain. Aspects related to instant messaging and social networks as the main channels for the transmission of fake news, the lack of credibility of public opinion towards the media in general, as well as the main disinformative contents, have been, among others, the most interesting contributions provided by this study on disinformation during the health crisis as result of COVID-19. 

We will now present the main conclusions of the study carried out, which provide answers to the research questions that have guided the work. At the same time, the discussion is carried out by relating these conclusions to state of the art research related to the topic addressed: -First, the results show an absolute interest on the part of the respondents in all the information related to COVID-19. Almost all the participants in the study have constantly monitored the content related to the pandemic. In this respect, as De las Heras et al. [46] pointed out, messages about coronavirus have been channelled mainly by the media and especially by generalist television stations. Thus, the main channels used to access the information have been television and the online press, as also concluded in the study by Montaña Blasco et al. [47], followed by WhatsApp and the social network Facebook, which stand out as the most used social networks to learn about the coronavirus [48].-With regard to the sources considered most reliable in obtaining information, the data show a significant lack of credibility of the media in general. None of the sources used by respondents have received high ratings of reliability; in this sense, and in line with the considerations of Masip et al. [1]. We are therefore faced with a scenario in which it is difficult to manage the population’s uncertainty when one of the main actors in the system, the media, does not have a sufficient reputation and is valued as an instrument of manipulation at the service, fundamentally, of the economic elites, as concluded in their study Villafañe et al. [49]. The sources that generate most credibility for information about coronavirus are the online press and television, followed by institutional websites. However, in addition to websites, the printed press and radio are considered the most reliable media, as they are the channels in which the dissemination of fake news related to the COVID-19 is lowest. It is worth noting the low reliability given to WhatsApp and the social network Facebook, as they are considered more likely to generate false news, despite being two of the main channels of access to information. This can happen because, as reflected in the latest results from Zuckerberg’s company [50], which not only owns Facebook but also WhatsApp, Messenger and Instagram, the majority of WhatsApp users also use Facebook. In turn, according to the Digital 2020 Global Digital Overview [51], social networks do not stop growing. In January 2020, Facebook had almost 2.449 million active users in one month while WhatsApp had more than 1.600 million active users in the same time. The enormous penetration of both Facebook and WhatsApp explains than citizens receive news through these networks related to the pandemic, news that can be true or, on the contrary can be fake.-With regard to the dissemination of fake news, it is worth noting the ease with which it spreads in these scenarios, characterised by a lack of leadership on the part of political representatives and the media. The majority of those surveyed stated that they had identified false news related to COVID-19, received mainly via instant messaging (WhatsApp) and the social network Facebook, recognising in them a negative intention that does not in any way benefit the achievement of a state of calm and understanding necessary in cases of collective panic. The most recurrent contents of the hoaxes that have been most widely disseminated have been those related to the different ways of preventing and detecting the spread of the virus, as well as the different medicines that can be taken or not taken in the presence of symptoms. Similarly, false news related to supposed statements or communications from health professionals and state security personnel has been prolific. The constant variations in the information issued by government bodies, sometimes with contradictory messages, have generated a situation of uncertainty in a society that has benefited from the entry of false news, through the different information channels.-As for citizens’ evaluation of institutional information on the pandemic, the data show some dissatisfaction in this regard, with 42% of respondents rating the information received negatively. This is a high percentage, if we consider that the political representatives, due to the service commitment acquired towards the citizens, have the duty to communicate with transparency, coherence and forcefulness, with the firm purpose of achieving the understanding of all. In this sense, Costa-Sánchez and López García [52] highlight the enormous responsibility of institutional actors and the media in communicating with citizens. As Xifra [24] points out, crisis situations are the time for public relations and management of reputational risk. In this sense, the study by Ibañez Peiró [53] pointed out in its conclusions the main errors made in institutional communication during the pandemic.-With regard to the negative effects of fake news on society, the vast majority of those surveyed attributed a high level of repercussion to the hoaxes related to COVID-19, considering the social impact generated by the resulting alarm situation to be serious or very serious. On the other hand, they also considered the negative consequences at a reputational level for the entities or people involved in the discourse to be important. In a hyperconnected society such as the one we live in, in which the generation and diffusion of content is fleeting, false news takes on an important dimension. In this sense, the majority of those surveyed expressed the need for legal regulation in this area.

Thus, as we have been pointing out in this Conclusions section, our study confirms and completes the findings of previous research. In turn, the health crisis that has paralysed normal activity in the political, economic, health, social and educational spheres has led to a complex situation of weariness and uncertainty, which requires effective communication by the main actors to enable society to understand, in a clear and reliable way, the seriousness of what is happening.

## Figures and Tables

**Figure 1 ijerph-18-01781-f001:**
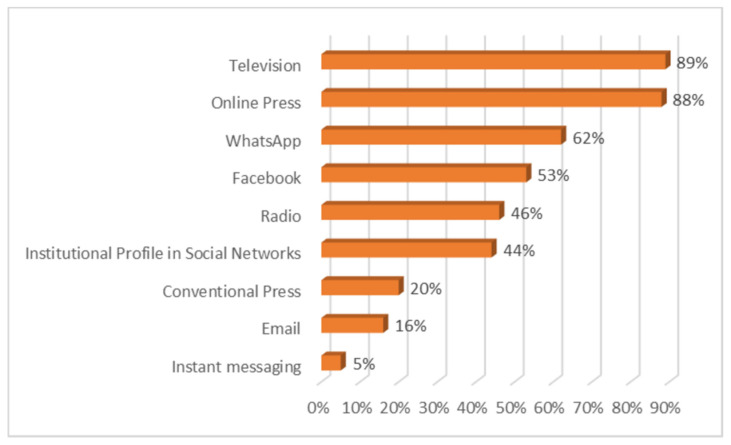
Main means used for information on COVID-19.

**Figure 2 ijerph-18-01781-f002:**
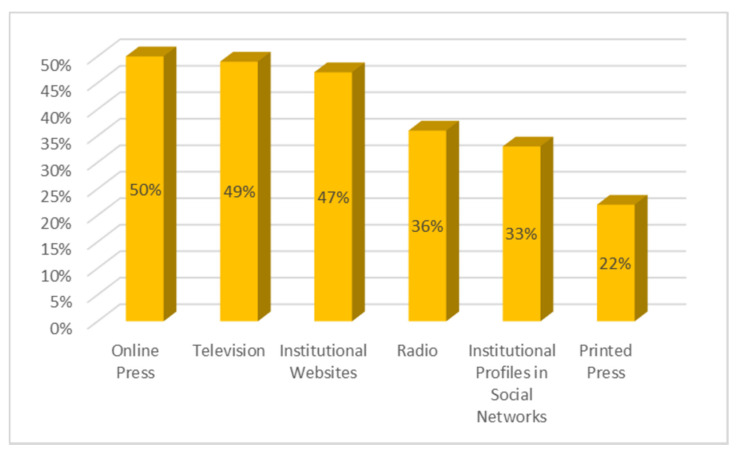
Sources that generate greater credibility for information on the COVID-19.

**Figure 3 ijerph-18-01781-f003:**
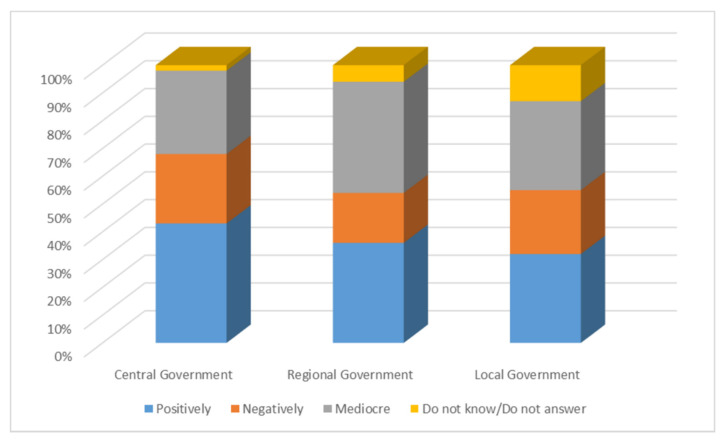
Comparison of the assessment of the information provided by central, regional and local governments on the COVID-19.

**Figure 4 ijerph-18-01781-f004:**
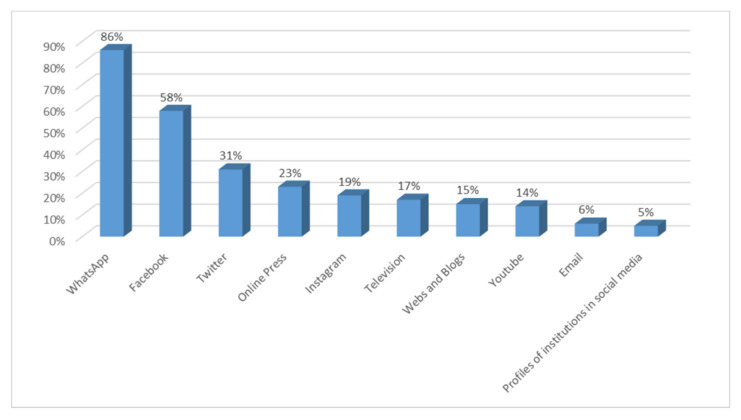
Main ways which citizens receive false news.

**Figure 5 ijerph-18-01781-f005:**
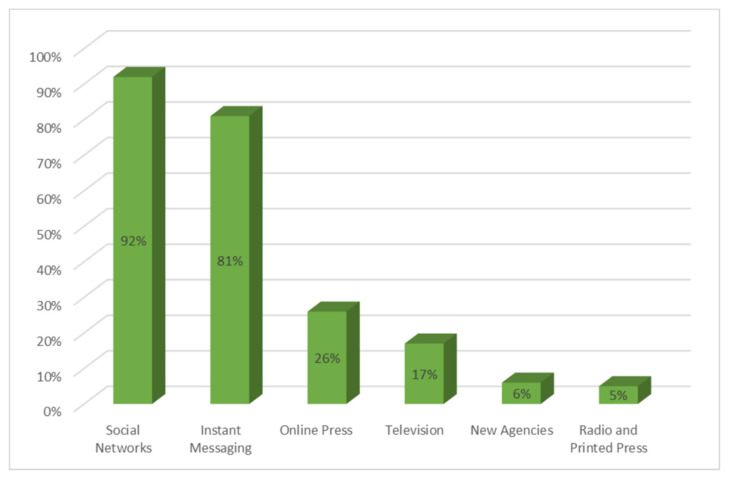
Media that citizens consider to have the highest amount of fake news.

**Figure 6 ijerph-18-01781-f006:**
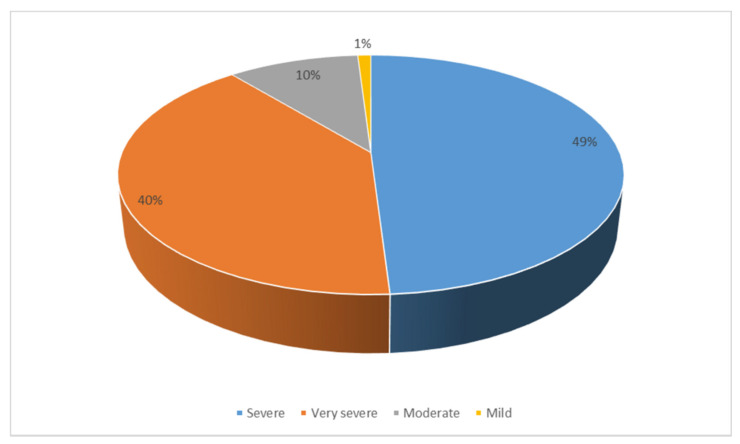
Degree of impact of fake news on society.

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
