# Peer review of "Infodemic and Fake News in Spain during the COVID-19 Pandemic"

_ijerph, 2021, doi:10.3390/ijerph18041781_

Round 1
Reviewer 1 Report
This paper has potential to make significant practical contributions. Below, I would like to suggest a few points that might be helpful for future revision.
- Line 47: A paragraph needs more than one sentence. Same in line 84, 109, 123, 168, 183, 234, 283, 318.
- Line 56: Direct citation needs a page number. Same in line 63, 81, 96, 101, 118, 126.
- Line 78: What does ‘strengthened communication’ mean? Mass and social media, suggested by the authors as key disseminators of fake news, also belong to the umbrella term ‘communication’. What kind of communication do the authors refer to that needs to be strengthened? What are the ways to strengthen communication?
- Line 97: The sentence is incomplete.
- Line 174: Higher education degree? Postgraduate degree?
- Figure 1: What does the ‘institutional profile in social networks’ category mean? Are they social network pages managed by institutions? Same in line 240, what does ‘profiles of government institutions’ indicate? Is it government-related website or a Facebook page managed by the government?
- Figure 2: Institutional websites (there is a typo)
- Figure 3: Negatively (there is a typo)
- Line 223: classify it as regular (Does it mean mediocre?). Same in Figure 3, is it ‘mediocre’ instead of regular?
- Figure 5: It is very interesting that most of the survey respondents believed that social networking services contain a very high number of fake news, while, at the same time, they used Facebook and WhatsApp quite heavily (53% and 62%) to get information on COVID-19 (according to Figure 1). How should we reconcile such discrepancy that people use social networking services for information while they believe they get the largest number of fake news from the social networking services? The level of credibility was rated quite low with the social networking services as well (according to Figure 2). As the authors directly mentioned in line 340, this phenomenon warrants further explanation.
Author Response
Thank you for your considerations and appreciations. They have enriched us a lot. Next, I will explain to you point by point the modifications made according to your instructions.
|
1.- Line 47: A paragraph needs more than one sentence. Same in line 84, 109, 123, 168, 183, 234, 283, 318. |
All these paragraphs have been modified (lines 47, 84, 109, 123, 168, 183, 234, 283 and 318), so that there are no longer single-sentence paragraphs.
|
|
2.- Line 56: Direct citation needs a page number. Same in line 63, 81, 96, 101, 118, 126 |
Pages have been included in all direct citations.
|
|
3.- Line 78: What does ‘strengthened communication’ mean? Mass and social media, suggested by the authors as key disseminators of fake news, also belong to the umbrella term ‘communication’. What kind of communication do the authors refer to that needs to be strengthened? What are the ways to strengthen communication? |
The explanation on how to strengthen communication has been added. By "strengthened communication" we understand that it is necessary to have a reinforced and strengthened communication, even more so in crisis situations in which communication management is of vital importance so that the crisis does not have a negative impact on the image or, failing that, to minimise its impact or to come out stronger. |
|
4.- Line 97: The sentence is incomplete. |
The sentence has been expanded to make it more complete and better understood. |
|
5.- Line 174: Higher education degree? Postgraduate degree? |
The term "Studies" has been replaced by "degree". |
|
6.- Figure 1: What does the ‘institutional profile in social networks’ category mean? Are they social network pages managed by institutions? Same in line 240, what does ‘profiles of government institutions’ indicate? Is it government-related website or a Facebook page managed by the government? |
We researchers wanted to say the following: - “Institutional profile in social networks” indicate social network pages managed by institutions. - “Profiles of government institutions” indicate government-related website. |
|
7.- Figure 2: Institutional websites (there is a typo) |
The typographical error in Figure 2 has been corrected. |
|
8.- Figure 3: Negatively (there is a typo) |
The typographical error in Figure 3 has been corrected. |
|
9.- Line 223: Classify it as regular (Does it mean mediocre?). Same in Figure 3, is it ‘mediocre’ instead of regular? |
The term "regular" has been replaced by "mediocre" both in the text and in Figure 3. |
|
10.- Figure 5: It is very interesting that most of the survey respondents believed that social networking services contain a very high number of fake news, while, at the same time, they used Facebook and WhatsApp quite heavily (53% and 62%) to get information on COVID-19 (according to Figure 1). How should we reconcile such discrepancy that people use social networking services for information while they believe they get the largest number of fake news from the social networking services? The level of credibility was rated quite low with the social networking services as well (according to Figure 2). As the authors directly mentioned in line 340, this phenomenon warrants further explanation. |
The results and conclusions reflect the fact that the majority of respondents believe that a large number of fake news items are spread through social networks and that they use both Facebook and WhatsApp for information purposes. Reports have been added to explain the possible reason for this:: Contrast the fact that a large part of respondents use WhatsApp and Facebook to find out about COVID-19 when there are other sources that generate a greater degree of credibility (Results). This can happen because, as reflected in the latest results forma Zuckerberg’s company [50], which not only owns Facebook but also WhatsApp, Messenger and Instagram, the majority of WhatsApp users also use Facebook. In turn, according to the Digital 2020 Global Digital Overview [51], social networks don’t stop growing. Thus, in January 2020, Facebook had almost 2.449 million active users in one month while WhatsApp has more than 1.600 million active users in the same time. The enormous penetration of both Facebook and WhatsApp explains than citizens receive news through these networks related to the pandemic, news that can be true or, on the contrary can be fake (Conclusions).
|
|
OTHER CLARIFICATIONS |
The references have been extended to 53. New references have been included in the state of the art and in the discussion:
- Catalan-Matamoros, D.; Elías, C. Vaccine Hesitancy in the Age of Coronavirus and Fake News: Analysis of Journalistic Sources in the Spanish Quality Press. Int. J. Environ. Res. Public Health 2020, 17(21), 8136. https://doi.org/10.3390/ijerph17218136 - McQueen, S. From Yellow Journalism to Tabloids to Clickbait: The Origins of Fake News in the United States. In Information Literacy and Libraries in the Age of Fake News; Agosto, D.E., Ed.; ABC-CLIO, LLC: Santa Barbara, CA, USA, 2018; pp. 12–36. - Duplaga, M. The Determinants of Conspiracy Beliefs Related to the COVID-19 Pandemic in a Nationally Representative Sample of Internet Users. Int. J. Environ. Res. Public Health 2020, 17(21), 7818. https://doi.org/10.3390/ijerph17217818 - Hernández-García, I.; Giménez-Júlvez, T. Information in Spanish on the Internet about the Prevention of COVID-19. Int. J. Environ. Res. Public Health 2020, 17(21), 8228. https://doi.org/10.3390/ijerph17218228 - Montaña Blasco, M.; Ollé Castellà, C; Lavilla Raso, M. Impacto de la pandemia de Covid-19 en el consumo de medios en España. Revista Latina de Comunicación Social, 78, 2020, 155-167. https://www.doi.org/10.4185/RLCS-20-1472 - López Rico, C.M.; González-Esteban, J.L.; Hernández-Martínez, A. Consumo de información en redes sociales durante la crisis de la COVID-19 en España. Revista de Comunicación y Salud, 10 (2), 2020, 461-481. https://doi.org/10.35669/rcys.2020.10(2).461-481 - Facebook Q1 2020 Results, 2020. https://s21.q4cdn.com/399680738/files/doc_financials/2020/q1/Q1-2020-FB-Earnings-Presentation.pdf - Digital 2020: Global Digital Overview, 2020. https://datareportal.com/reports/digital-2020-global-digital-overview - Ibáñez Peiró, A. La actividad informativa del Gobierno español durante la emergencia sanitaria provocada por el coronavirus, COVID-19. Revista Española de Comunicación en Salud, Suplemento 1, 2020. 304-318. https://doi.org/10.20318/rsc.2020.5441
Due to modifications, the number of lines has changed.
All revisions and reviewer considerations are highlighted in the text using the "Track Changes" function in Microsoft Word, so that changes are easily visible to the editors and reviewers. Reviewer is indicated and is derived to the cover letter.
|
Reviewer 2 Report
The topic is relevant, research questions are interesting and the answers to them can contribute to fake news research and help stop their spread. I very much hope that the following observations will help the authors to improve the manuscript and provide valuable insights for the scientific community.
The authors in the study should answer the following questions:
1. What was the method of selecting the survey respondents? Obviously, submitting a questionnaire on Google does not maintain the sampling steps. That causes some risks of biased responses. Nevertheless, such a method of sample selection is possible, but the method should be indicated correctly and based in detail on approved methodologies.
2. What was the method of data analysis? Why the demographic questions were included in the survey if they were not used in the presentation of the results. For example, it would be interesting to single out how responses differ by age group, education, or other. Providing answers to just one question greatly simplifies the study. This method of analysis does not provide enough information for the conclusions, especially since the study is not representative.
3. It is not clear how the respondents distinguish false news, so the answers show only their implied recognition of false news, dissemination and channels.
4. The scientific discussion is absent in the manuscript, so it is not clear how this study complements, refutes, or confirms similar research by other researchers. There has been a particularly large number of recent studies on the subject., therefore the comparison and the insights of the authors on that would be extremely useful.
Please take it as helpful suggestions for further studies as well as for this manuscript in order to contribute better to this important scientific topic. I wish you all the best!
Author Response
Thank you for your considerations and appreciations. They have enriched us a lot. Next, I will explain to you point by point the modifications made according to your instructions.
|
1.- What was the method of selecting the survey respondents? Obviously, submitting a questionnaire on Google does not maintain the sampling steps. That causes some risks of biased responses. Nevertheless, such a method of sample selection is possible, but the method should be indicated correctly and based in detail on approved methodologies. |
The survey was answered by a total of 1115 people and was active for two months. It was launched on 6 April 2020 and closed on 8 June of the same year and covered all of Spain. It coincided with the period of home confinement and the period of the toughest restrictions. The survey was carried out in Google Forms software. Regarding the selection of the sample, a simple random sampling was carried out with online collection methods, using the Internet as a collection tool. Thus, it was distributed online, especially through social networks and instant messaging such as WhatsApp. A correct design of the questionnaire was emphasized to avoid bias. Since the questionnaire was carried out randomly, identifying data was not processed and respondents were informed only of the purpose of the methodological tool and the study. To enhance an adequate control, we proceeded to calculate the sampling error of the survey which, in the present case, had a reduced margin of error (+_3) and a confidence level of 95%. The reliability of the survey is thus evident. The analysis of the data has been carried out through statistical calculations. |
|
2.- What was the method of data analysis? Why the demographic questions were included in the survey if they were not used in the presentation of the results. For example, it would be interesting to single out how responses differ by age group, education, or other. Providing answers to just one question greatly simplifies the study. This method of analysis does not provide enough information for the conclusions, especially since the study is not representative. |
The sociodemographic questions included in the questionnaire were only intended to describe the sample resulting from the dissemination of the self-administered questionnaires.
|
|
3.- It is not clear how the respondents distinguish false news, so the answers show only their implied recognition of false news, dissemination and channels |
This research has not investigated the criteria that citizens take into account to detect false news. Respondents were simply asked if they had identified fakes news. |
|
4.- The scientific discussion is absent in the manuscript, so it is not clear how this study complements, refutes, or confirms similar research by other researchers. There has been a particularly large number of recent studies on the subject., therefore the comparison and the insights of the authors on that would be extremely useful. Please take it as helpful suggestions for further studies as well as for this manuscript in order to contribute better to this important scientific topic. I wish you all the best! |
In the article there are a total of 33 sources with research related which are referenced both in the state of the question and in the conclusions, to contextualize our study and to compare the results obtained. However, we have slightly expanded the sources based on your suggestions. |
|
OTHER CLARIFICATIONS |
The references have been extended to 53. New references have been included in the state of the art and in the discussion:
- Catalan-Matamoros, D.; Elías, C. Vaccine Hesitancy in the Age of Coronavirus and Fake News: Analysis of Journalistic Sources in the Spanish Quality Press. Int. J. Environ. Res. Public Health 2020, 17(21), 8136. https://doi.org/10.3390/ijerph17218136 - McQueen, S. From Yellow Journalism to Tabloids to Clickbait: The Origins of Fake News in the United States. In Information Literacy and Libraries in the Age of Fake News; Agosto, D.E., Ed.; ABC-CLIO, LLC: Santa Barbara, CA, USA, 2018; pp. 12–36. - Duplaga, M. The Determinants of Conspiracy Beliefs Related to the COVID-19 Pandemic in a Nationally Representative Sample of Internet Users. Int. J. Environ. Res. Public Health 2020, 17(21), 7818. https://doi.org/10.3390/ijerph17217818 - Hernández-García, I.; Giménez-Júlvez, T. Information in Spanish on the Internet about the Prevention of COVID-19. Int. J. Environ. Res. Public Health 2020, 17(21), 8228. https://doi.org/10.3390/ijerph17218228 - Montaña Blasco, M.; Ollé Castellà, C; Lavilla Raso, M. Impacto de la pandemia de Covid-19 en el consumo de medios en España. Revista Latina de Comunicación Social, 78, 2020, 155-167. https://www.doi.org/10.4185/RLCS-20-1472 - López Rico, C.M.; González-Esteban, J.L.; Hernández-Martínez, A. Consumo de información en redes sociales durante la crisis de la COVID-19 en España. Revista de Comunicación y Salud, 10 (2), 2020, 461-481. https://doi.org/10.35669/rcys.2020.10(2).461-481 - Facebook Q1 2020 Results, 2020. https://s21.q4cdn.com/399680738/files/doc_financials/2020/q1/Q1-2020-FB-Earnings-Presentation.pdf - Digital 2020: Global Digital Overview, 2020. https://datareportal.com/reports/digital-2020-global-digital-overview - Ibáñez Peiró, A. La actividad informativa del Gobierno español durante la emergencia sanitaria provocada por el coronavirus, COVID-19. Revista Española de Comunicación en Salud, Suplemento 1, 2020. 304-318. https://doi.org/10.20318/rsc.2020.5441
Due to modifications, the number of lines has changed.
All revisions and reviewer considerations are highlighted in the text using the "Track Changes" function in Microsoft Word, so that changes are easily visible to the editors and reviewers. Reviewer is indicated and is derived to the cover letter.
|
Reviewer 3 Report
This is a very interesting paper concerning perception of COVID-19 news and information credibility aspects among the public in Spain. It potentially brings some new aspects to the table, if the paper can be better situated among the existing state of the art on the topic.
The term infodemic was not introduced during the COVID-19 pandemic, it would be more accurate to say that the term has become popularised currently rather than invented. Also the reference to WMDs and the Gulf War, should be amended to the Iraq War (2003 as opposed to the 1990-91 Gulf War).
One thing missing from the manuscript is a comprehensive overview of the existing state of the art literature on aspects of fake news and infodemic and its impact upon the crisis caused by COVID-19. These are key points to this paper and should be developed further in order to bring context and better understanding how this study fits into the bigger picture and greater context to the results of the public survey.
Author Response
Thank you for your considerations and appreciations. They have enriched us a lot. Next, I will explain to you point by point the modifications made according to your instructions.
|
This is a very interesting paper concerning perception of COVID-19 news and information credibility aspects among the public in Spain. It potentially brings some new aspects to the table, if the paper can be better situated among the existing state of the art on the topic. The term infodemic was not introduced during the COVID-19 pandemic, it would be more accurate to say that the term has become popularised currently rather than invented. Also the reference to WMDs and the Gulf War, should be amended to the Iraq War (2003 as opposed to the 1990-91 Gulf War). One thing missing from the manuscript is a comprehensive overview of the existing state of the art literature on aspects of fake news and infodemic and its impact upon the crisis caused by COVID-19. These are key points to this paper and should be developed further in order to bring context and better understanding how this study fits into the bigger picture and greater context to the results of the public survey |
Following the reviewer's indications, it has been clarified that the term infodemic is not new, but has become popular during the crisis. The Gulf War has also been replaced by the Iraq War. Likewise, recent publications have been included, avant-garde literature that completes the state of the art. In total, the theoretical framework has 44 different sources, most of them from 2019 and 2020. |
|
OTHER CLARIFICATIONS |
The references have been extended to 53. New references have been included in the state of the art and in the discussion:
- Catalan-Matamoros, D.; Elías, C. Vaccine Hesitancy in the Age of Coronavirus and Fake News: Analysis of Journalistic Sources in the Spanish Quality Press. Int. J. Environ. Res. Public Health 2020, 17(21), 8136. https://doi.org/10.3390/ijerph17218136 - McQueen, S. From Yellow Journalism to Tabloids to Clickbait: The Origins of Fake News in the United States. In Information Literacy and Libraries in the Age of Fake News; Agosto, D.E., Ed.; ABC-CLIO, LLC: Santa Barbara, CA, USA, 2018; pp. 12–36. - Duplaga, M. The Determinants of Conspiracy Beliefs Related to the COVID-19 Pandemic in a Nationally Representative Sample of Internet Users. Int. J. Environ. Res. Public Health 2020, 17(21), 7818. https://doi.org/10.3390/ijerph17217818 - Hernández-García, I.; Giménez-Júlvez, T. Information in Spanish on the Internet about the Prevention of COVID-19. Int. J. Environ. Res. Public Health 2020, 17(21), 8228. https://doi.org/10.3390/ijerph17218228 - Montaña Blasco, M.; Ollé Castellà, C; Lavilla Raso, M. Impacto de la pandemia de Covid-19 en el consumo de medios en España. Revista Latina de Comunicación Social, 78, 2020, 155-167. https://www.doi.org/10.4185/RLCS-20-1472 - López Rico, C.M.; González-Esteban, J.L.; Hernández-Martínez, A. Consumo de información en redes sociales durante la crisis de la COVID-19 en España. Revista de Comunicación y Salud, 10 (2), 2020, 461-481. https://doi.org/10.35669/rcys.2020.10(2).461-481 - Facebook Q1 2020 Results, 2020. https://s21.q4cdn.com/399680738/files/doc_financials/2020/q1/Q1-2020-FB-Earnings-Presentation.pdf - Digital 2020: Global Digital Overview, 2020. https://datareportal.com/reports/digital-2020-global-digital-overview - Ibáñez Peiró, A. La actividad informativa del Gobierno español durante la emergencia sanitaria provocada por el coronavirus, COVID-19. Revista Española de Comunicación en Salud, Suplemento 1, 2020. 304-318. https://doi.org/10.20318/rsc.2020.5441
Due to modifications, the number of lines has changed.
All revisions and reviewer considerations are highlighted in the text using the "Track Changes" function in Microsoft Word, so that changes are easily visible to the editors and reviewers. Reviewer is indicated and is derived to the cover letter.
|
Round 2
Reviewer 2 Report
Dear Authors,
thank you for your answers and the improvements. However, the scientific article should have a purpose to contribute to the field. The scientific discussion is still absent in the manuscript, so it is not clear how this study complements, refutes, or confirms similar research by other researchers and how important are the conclusions.
The data are presented in the simplest way. The analytical approach is necessary to present the survey data.
Best regards,
Author Response
|
Dear Authors,
thank you for your answers and the improvements. However, the scientific article should have a purpose to contribute to the field. The scientific discussion is still absent in the manuscript, so it is not clear how this study complements, refutes, or confirms similar research by other researchers and how important are the conclusions.
The data are presented in the simplest way. The analytical approach is necessary to present the survey data.
|
Dear reviewer,
Thank you very much for your recommendations. They have been taken into account and included in the article:
- At the beginning of the Discussion and Conclusions the main contributions have been presented. - In Discussion and Conclusions, the contributions of our research have been compared, refuted or confirmed with those of other authors. Specifically, eight scientific publications have been used (Xifra; Masip, Aran-Ramspott, Ruiz-Caballero, Suau, Almenar, Puertas-Graell; De las Heras-Pedrosa, Sánchez Núñez, Peláez; Montaña Blasco, Ollé Castellà, Lavilla Raso; López Rico, González-Esteban, Hernández-Martínez; Villafañe, Ortiz-de-Guinea-Ayala; Martín-Sáez; Costa-Sánchez, López García; Ibáñez Peiró) and two reports (Facebook Q1 2020 Result Reports; Digital 2020: Global Digital Overview) - At the end of Discussion and Conclusions it has been emphasised how the research confirms and complements similar research - The word Discussion has been added to the heading, so that it is now called Discussion and Conclusions.
The results obtained in this research are relevant and represent a contribution to research on a topic of great social interest, such as fakes news. The study carried out has made it possible to achieve the objectives set out in the article.
|
Reviewer 3 Report
All of the comments and suggestions from the first round have been completed to an acceptable standard.
Author Response
Thank you for your consideration.